# Analysis of the Anaerobic Power Output, Dynamic Stability, Lower Limb Strength, and Power of Elite Soccer Players Based on Their Field Position

**DOI:** 10.3390/healthcare10112256

**Published:** 2022-11-10

**Authors:** Ali AlTaweel, Shibili Nuhmani, Mohammad Ahsan, Wafa Hashem Al Muslem, Turki Abualait, Qassim Ibrahim Muaidi

**Affiliations:** Department of Physical Therapy, College of Applied Medical Sciences, Imam Abdulrahman Bin Faisal University, Dammam 31451, Saudi Arabia

**Keywords:** physical fitness, functional test, star excursion balance test, single leg vertical jump, triple hop for distance

## Abstract

Soccer players require a high degree of aerobic and anaerobic fitness to perform well throughout the game as per their position in the field. This study aimed to investigate the differences in anaerobic power output, dynamic stability, lower limb strength, and power among elite soccer players based on their field position. A cross-sectional population of 96 elite soccer players with average age 23.10 ± 4.35 years, weight 69.99 ± 9.71 kg, height 174.84 ± 6.64 cm, and body mass index 22.84 ± 2.39 kg/m^2^ from various soccer clubs in Saudi Arabia was tested for their anaerobic power output, dynamic stability, lower limb strength, and power performance. All the participants have more than 4 years of experience in competitive soccer events. Tests included a measure of single-leg vertical jump, star excursion balance test, and single-leg triple hop test for distance. The players were divided into four groups (goalkeepers, defenders, midfielders, and attackers) based on their self-reported position on the field. One-way ANOVA was used to determine the differences between all variables according to the players’ position. In addition, partial eta-squared (ηp2) was used to report effect sizes. The results revealed significant differences between positions in the anaerobic power output (*p* = 0.012, ηp2 = 0.312), dynamic stability {Anterior (*p* = 0.004, ηp2 = 0.235), Anteromedial (*p* = 0.007, ηp2 = 0.622), Anterolateral (*p* = 0.011, ηp2 = 0.114)}, and lower limb strength, and power (*p* = 0.008, ηp2 = 0.421). At the same time, goalkeepers’ performance was significantly superior to midfielders (*p* = 0.006) in the anaerobic power output. In addition, lower limb strength and power was significantly higher (*p* = 0.004) for goalkeepers than for midfielders, with a similar trend in dynamic stability (*p* = 0.007). These results exhibited differences in anaerobic power output, dynamic stability, lower limb strength, and power performance based on the players’ positions. The investigation may assist the practitioner in designing training programs for the players according to their position for performance improvement.

## 1. Introduction

Soccer is one of the most popular sports around the world. To participate in soccer, a player performs high-intensity technical and tactical activities such as sprinting, dribbling, kicking, jumping, dodging, tackling, feinting, heading the ball, and changing ball direction [1]. In addition to these skills, soccer players should improve their movements without the ball in all field positions. Soccer players have their positions on the field and deploy strategies and tactics during the game [2]. This requires the player to have the physical ability to put in the necessary effort and his best effort [3]. Coaches usually design the training to help the players attain their optimal performance to accomplish this. They incorporate various components of the training, depending on the player’s position or on the tactic of the game, which can influence the performance. Training based on the players’ positions is recognized as one of the best ways to improve the players’ performance and has become one of the main strategies for preparing players for soccer matches [4].

Soccer games require a high degree of aerobic and anaerobic fitness to participate. Players must be able to conduct repeated maximal or sub-maximal demanding tasks that need a high level of oxidative energy throughout the game [5]. Most sprints in games last approximately 10 s and cover a distance of 5 to 30 m. Maccurdy et al. identified that explosive movement such as a vertical jump is a key predictor of anaerobic power output. Anaerobic power output is responsible for more than half of the variance in performance measures [6]. A single leg vertical jump is an example of anaerobic power. In single leg vertical jumps, male football players have recorded mean values between 44 and 60 cm [7].

Dynamic stability is not only necessary for sports participation, but it can also be used to predict injury [8]. Dynamic stability impairments have been extensively studied as a predictor of lower extremity injuries in soccer players [9]. The star excursion balance test is a common, reliable, and non-invasive test to measure dynamic stability. It has been pro-posed that utilizing an injury screening technique that assesses dynamic stability performance can help identify soccer players who are at risk of injury [10]. The position of soccer players was associated with greater dynamic stability [11]. Pau et al. suggested that soccer players’ ability of dynamic stability is a critical aspect in obtaining optimum performance and avoiding injury risk [12].

One of the most frequently used functional performance test measures is the triple hop for distance, which is used to determine lower limb strength and power. Lower limb strength and power are essential physical attributes in soccer, and soccer players’ high-intensity endurance capacity may predict future career success [13]. Hamilton et al. reported that the triple hop for distance is a strong predictor of lower-limb muscle strength and power in a healthy soccer population, implying that it might be used as a preseason screening test [14].

The player’s positions on the soccer field are classified by denomination and function. A wide range of studies available in the literature has covered soccer players according to their playing position [15,16,17,18,19] and investigated their physical characteristics [20]. Even though a few studies investigated some of the physical characteristics of soccer players in the Middle East region, none of them investigated the most important soccer performance characteristics, such as anaerobic power, dynamic stability, and the strength and power of lower leg muscles based on the field position in this region. Moreover, this is the first study that investigated differences in performance characteristics based on field position among Saudi soccer players. Ethnicity and genetics have a major role in determining athletes’ physical, physiological characteristics, and performance. It has been reported in previous studies that the athletes from the middle east region differ from other parts of the world in physical and anthropometric characteristics [21].

This study aims to examine the differences in anaerobic power output, dynamic stability, lower limb strength, and power performance based on player positions. Recognizing these differences is important and may help understand the requirements of each player to reach their best performance in their position. This might help coaches and trainers to design appropriate training programs for individuals and groups of players based on their positions. Furthermore, it may explain the factors that reduce player performance and the physical characteristics that contribute to success in a position. In addition, it may help coaches and medical teams design appropriate training to prevent player injuries in their positions on the field. The study hypotheses that there will be significant differences for anaerobic power output, dynamic stability, lower limb strength, and power performance among positions of elite soccer players.

## 2. Materials and Methods

### 2.1. Study Design and Setting

A cross-sectional study design was adopted to conduct this study. This study was conducted in the eastern province of Saudi Arabia. 

### 2.2. Sample Size

The sample size calculation is based on a previous study [22] for comparing means and the approach followed as probability of stating a difference when no difference exists (95%), standard deviation of sample (7.3), and difference between population mean and sample mean (4.3). The calculated sample size was 21 participants for each field position group was increased by 12.5% to reduce sampling errors. All calculations have been made according to the formula proposed as sample sizes for comparative research studies [23]. Therefore, a total of 96 participants (24 for each position) were recruited. 

### 2.3. Ethical Approval

An ethical approval was obtained from the Institute Review Board of the University of Imam Abdulrahman Bin Faisal (IRB-PCS-2018-03-011). Each participant provided written informed consent prior to participation in the test. 

### 2.4. Participants

The volunteer participants were 96 male elite soccer players from the eastern province of Saudi Arabia (24 from each group) for four field position groups, i.e., goalkeepers, defenders, midfielders, and attackers. All the participants have more than 4 years of experience in competitive soccer events. All the participants were recruited from the various soccer clubs who participate in the Saudi professional league, which is the top division league in the Saudi Arabian soccer league system. Participants with a history of musculoskeletal injury in the lower limb and back in the past three months, any neurological problems, systemic diseases, use of any medication, and any biomechanical abnormalities that affect the performance were excluded from the study. 

### 2.5. Procedure

All the participants were screened in the physiotherapy lab at Imam Abdulrahman bin Faisal University based on the inclusion and exclusion criteria. The procedure, associated benefits, and relative risks of the study were explained to the participants prior to the actual tests. The players were asked about their position on the soccer field and their training experience. The detection of the dominant leg was recognized by the player’s self-reporting of their preferred leg. The anthropometric data of players were collected, including age, weight, height, and body mass index (BMI). The data were collected on the morning session between 8 AM to 11 AM. Anaerobic power output, dynamic stability, and lower limb strength and power of the soccer players were determined by the below-mentioned test and procedures. 

### 2.6. Outcome Measurements

#### 2.6.1. Anaerobic Power Output

The anaerobic output was determined by the Single Leg Vertical Jump (SLVJ) test using the vertec device (Jump USA 22550). The participant performed a single leg vertical jump on a flat non-slip surface. The participant wore a running shoe during the test. Participants stood away from the device at a distance of 15 cm in an upright position where both hands could be elevated to the level of the head to avoid any countermovement. They were asked to balance on the dominant leg while their knee was maintained at 90 degrees. Then, the participant jumped to their highest ability to touch the vane. Then, the investigator measured the highest point that the participant reached and calculated the differences between the standing reach and the jumping height. As performing the test allowed for only straight down and straight up movements, the participant was not allowed to perform a shuffle step, side-step, drop-step, and step allowed [24]. Three attempts were performed with a one minute rest between each attempt. The average of the three attempts was taken for analysis. Five minutes rest was given to the participant between each functional performance test [25]. 

#### 2.6.2. Dynamic Stability

Dynamic stability was measured by the star excursion balance test (SEBT). The test consists of four lines drawn on a layout. Two lines are made vertically and horizontally and perpendicular to each other. At 45 degrees of the perpendicular line, another two lines drawn were made vertically and horizontally and perpendicular to each other. The final shape showed a star with a 45 degree angle between all lines [26]. The participants stood on the dominant leg, and they received verbal and visual instruction from the examiner to begin the SEBT. They were asked to reach in the anterior medial, left anterolateral, left mediolateral, left posteromedial, posteromedial, right posteromedial, right mediolateral and right anterolateral directions by the other leg to reach the farthest distance possible with the toe. The participants kept their foot on the ground and maintained their balance while performing the test. The test was repeated if (1) the participant was not able to keep a single-limb stance, (2) the heel on the stance foot was elevated from the ground, (3) the weight shifted to the reaching foot in any of the three directions, or (4) the reach foot did not return to the starting position [27]. The participant performed three attempts with a one minute rest between the attempts and the average was calculated for data analysis. Five minutes rest was given between each functional performance tests. The data were normalized by leg length of the participants. The SEBT’s intraratar and interrater reliability (ICC) values for all directions ranged from 0.88 to 0.96, and from 0.83 to 0.93, respectively [28]

#### 2.6.3. Lower Limb Strength and Power

Lower limb strength and power was investigated by the triple hop for distance (THD). A horizontal line was marked as a starting line and a perpendicular line of 7 m distance was drawn by a tape. The participant stood on the dominant leg and the toe (edge of the shoes) on the predetermined line on the floor. The participant hopped forward as far as possible, jumping three times and landing on the same leg. They were able to move their arms during winging, but were asked to maintain their foot contact on the floor upon landing. The investigator measured the horizontal distance in centimeters from the heel starting position to the heel landing mark by a standard tape measurement [29]. The attempts were performed with one minute rest in between the attempts and the average horizontal displacement was calculated and taken for analysis [30]. The test–retest reliability of THD has been demonstrated as having an intraclass correlation coefficient = 0.98 in previous research [31].

### 2.7. Statistics Analysis

Descriptive and interferential statistics are provided in the form of means and standard deviations (Std). All the statistical analyses were done by IBM SPSS program software version 21.0. on windows. Normality and homogeneity of the data were confirmed by using Shapiro–Wilk’s test and Levene’s Test of Equality of Error Variances (*p* > 0.05), respectively. One-way ANOVA was used to determine the differences between all variables, including anaerobic power output, dynamic stability, lower limb strength, and power of the soccer players performance variables according to the players’ position. Multiple comparisons were made by using the Tukey post-hoc test if there were significant differences. In addition, partial eta-squared (ηp2) was used to report effect sizes as an ηp2 = 0.01 is considered a small effect; ηp2 = 0.06 is considered a medium effect; and ηp2 = 0.14 is considered a large effect [32]. The level of significance was set at *p* < 0.05 and confidence of intervals at 95%.

## 3. Results

Table 1 shows that the goalkeepers were more heavier in all outfield positions than the other groups (76.79 ± 12.38, *p* = 0.001). The goalkeepers are taller (178.42 ± 5.66) than the defenders (173.13 ± 7.11) and midfielders (172.63 ± 5.06); (*p* = 0.008). In the BMI, the result shows that the goalkeepers (24.02 ± 2.98, *p* = 0.043) are higher than all outfield positions.

Table 2 shows a comparison between elite soccer players for anaerobic power output according to their different player positions as significant at (*p* = 0.012, ηp2 = 0.312). The anaerobic power output results showed that there was a significant difference between groups in anaerobic power output.

Table 3 shows that only goalkeepers and midfielders have significant differences, whereas no significant differences were found among other players for anaerobic power output as per the playing positions.

The dynamic stability shows significant differences between places, as seen in Table 4. As the values indicate the mean scores, there were significant differences in dynamic stability—anterior (*p* = 0.014, ηp2 = 0.182) and anteromedial (*p* = 0.035, ηp2 = 0.169), but no differences were identified in other directions. 

Table 5 shows that the goalkeepers’ group significantly differed in anterior stability (*p* = 0.038 and *p* = 0.047) with defenders and midfielders, and goalkeepers significantly differed in anteromedial reach with defenders (*p* = 0.032) and midfielders (*p* = 0.024). Results show the significance differences in the mean scores between the groups.

Table 6’s findings revealed that there were differences in the lower limb strength and power of soccer players for their respective playing positions that were statistically signif-icant (*p* = 0.008, ηp2 = 0.421). Although effect size was determined as large effect size for lower limb strength and power between all groups.

The Table 7 demonstrated that only goalkeepers and midfielders had a significant difference (*p* = 0.004), whereas other players do not show any significant differences for lower limb strength and power.

## 4. Discussion

This study’s aims were to examine the differences in anaerobic power output, dynamic stability, lower limb strength, and power performance based on player positions. Our study showed differences in functional performance according to their playing positions. The anthropometric data of the players showed that goalkeepers were significantly heavier than all the players in the outfield positions. Goalkeepers were also significantly taller than defenders and midfielders. The result also showed that goalkeepers were significantly higher in BMI than all outfield players. Effect size showed age has a small effect size, BMI and height have a medium effect size, whereas weight has a large effect size. Goalkeepers’ performance was significantly superior to midfielders in anaerobic power, and goalkeepers were significantly superior in dynamic stability as measured by SEBT in certain directions. Goalkeepers had a significantly higher anterior and anteromedial reach than defenders and midfielders. Finally, the result showed that goalkeepers’ performance was significantly better than midfielders’ in the lower limb strength and power. 

In the present study, the result showed that in single-leg vertical jump performance, goalkeepers performed significantly better than midfielders. Single-leg vertical jump performance also showed a large effect size among different groups. Determining power output is essential because most activities and skills clearly benefit from having a larger anaerobic power threshold. Our results agree with a previous study conducted in Croatia, which found that goalkeepers were significantly better than other players in vertical jump performance [33]. Conversely, our results disagree with studies that showed that there was no significant difference between goalkeepers, defenders, midfielders, and attackers in vertical jumping performance [20,22,34,35]. Other studies of vertical jump performance have found that forward-positioned players are better than midfielders and fullbacks [33], and mid-fielders showed significantly lower jump height values than other positions [36]. In contrast, other studies found no significant differences in vertical jump performance between the outfielder positions [37]. 

In SEBT measurement, the anterior stability, medial stability, posteromedial stability, posterior stability, posterolateral stability, lateral stability, and anterolateral stability have a small effect size, whereas antromedial stability has a large effect size. There are few studies available in the literature that investigated the difference in dynamic balance according to the playing position of soccer players [36]. Pau, Ibba, et al. investigated the static balance in elite soccer players according to their positions, and the found that midfielders had significantly lower values in the sway area than defenders, which is in agreement with our study result [12]. Ates et al., 2019 investigated the dynamic balance performance of Turkish players according to their positions and reported no differences in playing position using the Y balance test [36]. The study results disagree with our study result, which shows that the goalkeepers in SEBT were different in many respects. Goalkeepers have longer reach than defenders in anterior, anterolateral, and anteromedial directions, and are higher than midfielders in anterior and anterolateral directions. Goalkeepers usually have more challenges in contacting opposing players during games, which may improve their dynamic balance better than those of the outfield players, especially in certain directions [38]. Dynamic stability of trained athletes improves as a result of their flexibility and joint range of motion, which contributes to a more flexible and stable chain of skillful movements and helps to protect them from injuries [8].

Our study showed that goalkeepers were higher than midfielders in the triple hop for distance jump, but there were no differences between the defenders, midfielders, and attackers. While performing the triple hop for distance, the players are required to move their center of mass in the horizontal direction [39], which influences dynamic stability. A large effect size was found for lower limb strength and power between all groups The goalkeepers in our study were higher in the anterior reach in the SEBT in different directions, which may explain the goalkeeper’s superiority in the triple hop jump for distance over the midfield players. Dynamic stability and lower leg strength positively contribute to jump performance and power generation [40]. Again, the similarity of the results in triple hop for distance jump between defenders, midfielders, and attackers might be because of the similarity in physical fitness and similar training received.

The present study has its limitations. First, the measurement was taken from dominant leg of the players whereas the non-dominant leg was not included in the analysis. Second, there are other factors such as hours of training, history, playing minutes, and playing tactics that were also not included in the analysis. Third, the study did not investigate those players who may play in different positions due to an absence of another player, which may influence the results. Fourth, our study players are divided into four positions: goalkeepers, defenders, midfielders and attackers. However, the players can be divided into more than four positions, such as the full back, external midfielders, etc. Fifth, the study did not include female soccer player; therefore, the results cannot be generalized to both genders of players. Furthermore, to find out the differences in anaerobic power output, dynamic stability, lower limb strength, and power performance, players’ ages and more types of other functional performance testing that determine the appropriate lower limbs functional test can be performed.

## 5. Conclusions

Our study showed significant differences in anaerobic power output, dynamic stability, and lower limb strength and power performance according to their playing position. The study’s hypotheses were accepted. Goalkeepers were more stable and significantly higher than midfielders in dynamic stability, and were higher than defenders and mid-fielders in dynamic stability. The study may help coaches and trainers to recognize the strengths and weaknesses of their players and design training programs to improve the weaker components and improve the performance of players in different playing positions and to design appropriate training programs for individuals and groups of players based on the players’ position.

## Figures and Tables

**Table 1 healthcare-10-02256-t001:** Anthropometric characteristics and analysis of variance for the elite soccer players ac-cording to their different playing positions.

Variables	Positions	Mean ± Std	95% CI Lower to Upper Limit	Partial Eta- Squared	*p* Value
**Age** **(years)**	Goalkeepers	23.58 ± 3.69	22.02–25.14	0.006	0.904
	Defenders	22.63 ± 3.57	21.12–24.13		
	Midfielders	23.08 ± 5.18	20.89–25.27		
	Attackers	23.13 ± 4.96	21.03–25.22		
**Weight (kg)**	Goalkeepers	76.79 ± 12.38	71.57–82.02	0.170	0.001 *
	Defenders	67.82 ± 7.38	64.70–70.93		
	Midfielders	66.68 ± 6.04	64.13–69.23		
	Attackers	68.70 ± 8.91	64.94–72.46		
**Height (cm)**	Goalkeepers	178.42 ± 5.66	176.03–180.81	0.119	0.008 *
	Defenders	173.13 ± 7.11	170.12–176.13		
	Midfielders	172.63 ± 5.06	170.49–174.76		
	Attackers	175.21 ± 7.23	172.16–178.26		
**BMI (kg/m^2^)**	Goalkeepers	24.02 ± 2.98	22.76–25.28	0.084	0.043 *
	Defenders	22.65 ± 2.32	21.67–23.63		
	Midfielders	22.38 ± 1.83	21.60–23.15		
	Attackers	22.32 ± 2.04	21.46–23.18		

* Significant at 0.05 level.

**Table 2 healthcare-10-02256-t002:** Statistical analysis of variance of anaerobic output values of soccer players for their playing position.

Variables (cm)	Positions	Mean ± Std	95% CI	Partial Eta- Squared	*p* Value
			Lower to Upper Limit		
Anaerobic Power Output	Goalkeepers	16.32 ± 2.29	15.35–17.28	0.312	0.012 *
	Defenders	15.38 ± 2.55	14.30–16.46		
	Midfielders	14.12 ± 1.75	13.38–14.86		
	Attackers	15.10 ± 2.32	14.12–16.07		

* Significant at 0.05 level.

**Table 3 healthcare-10-02256-t003:** Post-Hoc (Tukey) analysis of anaerobic output values of soccer players for their playing position.

Dependent Variable (cm)	Group	Group	Mean Difference	Std. Error	Effect Size %	*p* Value
Anaerobic Power Output	Goalkeepers	Defenders	0.94	0.65	26.9	0.476
		Midfielders	2.20 *	0.65	96.2	0.006 *
		Attackers	1.22	0.65	45.0	0.242
	Defenders	Goalkeeper	−0.94	0.65	26.9	0.476
		Midfielders	1.26	0.65	51.4	0.218
		Attackers	0.29	0.65	5.9	0.971
	Midfielders	Goalkeeper	−2.20 *	0.65	96.2	0.006 *
		Defenders	−1.26	0.65	51.4	0.218
		Attackers	−0.97	0.65	37.9	0.440
	Attackers	Goalkeepers	−1.22	0.65	45.0	0.242
		Defenders	−0.29	0.65	5.9	0.971
		Midfielders	0.97	0.65	37.9	0.440

* Significant at 0.05 level.

**Table 4 healthcare-10-02256-t004:** Statistical analysis of variance of dynamic stability values of soccer players for their playing position.

Dynamic Stability (cm)	Positions	N	Mean ± Std. Deviation	95% CI Lower to Upper Limits	Partial Eta- Squared	*p* Value
Anterior Stability	Goalkeepers	24	48.23 ± 5.42	45.94–50.52	0.082	0.014 *
	Defenders	24	44.58 ± 4.41	42.72–46.44		
	Midfielders	24	45.47 ± 4.43	43.60–47.34		
	Attackers	24	47.63 ± 5.96	45.11–50.15		
Anterioromedial Stability	Goalkeepers	24	50.95 ± 5.31	48.71–53.19	0.669	0.035 *
	Defenders	24	47.31 ± 4.53	45.39–49.22		
	Midfielders	24	48.61±4.44	46.73–50.48		
	Attackers	24	49.50 ± 5.51	47.18–51.83		
Medial Stability	Goalkeeper	24	51.44 ± 5.66	49.04–53.83	0.029	0.426
	Defenders	24	48.74 ± 5.32	46.49–50.99		
	Midfielders	24	50.30 ± 5.01	48.18–52.41		
	Attackers	24	50.29 ± 6.32	47.63–52.96		
Poteriomedial Stability	Goalkeepers	24	51.18 ± 5.89	48.69–53.67	0.027	0.476
	Defenders	24	48.66 ± 7.73	45.40–51.93		
	Midfielders	24	50.81 ± 4.33	48.98–52.64		
	Attackers	24	50.85 ± 6.27	48.20–53.49		
Posterior Stability	Goalkeepers	24	49.88 ± 6.79	47.01–52.75	0.003	0.960
	Defenders	24	49.09 ± 4.36	47.25–50.93		
	Midfielders	24	49.88 ± 5.45	47.58–52.19		
	Attackers	24	49.65 ± 6.42	46.94–52.36		
Posteriolateral Stability	Goalkeepers	24	47.76 ± 5.92	45.26–50.26	0.016	0.687
	Defenders	24	46.29 ± 5.08	44.14–48.43		
	Midfielders	24	47.30 ± 4.58	45.36–49.23		
	Attackers	24	46.11 ± 6.41	43.41–48.82		
Lateral Stability	Goalkeepers	24	42.69 ± 5.72	40.27–45.10	0.016	0.677
	Defenders	24	41.23 ± 4.34	39.40–43.07		
	Midfielders	24	42.53 ± 4.63	40.57–44.48		
	Attackers	24	41.25 ± 6.71	38.41–44.09		
Anteriolateral Stability	Goalkeepers	24	45.59 ± 5.00	43.48–47.70	0.059	0.126
	Defenders	24	42.40 ± 3.56	40.90–43.90		
	Midfielders	24	42.86 ± 5.63	40.48–45.24		
	Attackers	24	43.20 ± 5.47	40.89–45.51		

* Significant at 0.05 level.

**Table 5 healthcare-10-02256-t005:** Post-Hoc (Tukey) analysis of dynamic stability values of soccer players for their playing position.

Dependent Variable (cm)	Group	Group	Mean Difference	Std. Error	Effect Size %	*p* Value
Anterior Stability	Goalkeepers	Defenders	3.65	1.47	72.5	0.038 *
		Midfielders	2.76	1.47	48.9	0.047 *
		Attackers	0.60	1.47	9.8	0.677
	Defenders	Goalkeeper	−3.65	1.47	72.4	0.038 *
		Midfielders	−0.89	1.47	38.1	0.429
		Attackers	−3.05	1.47	81.1	0.056
	Midfielders	Goalkeeper	−2.76	1.47	48.9	0.047 *
		Defenders	0.89	1.47	38.1	0.429
		Attackers	−2.16	1.47	9.8	0.462
	Attackers	Goalkeepers	−0.60	1.47	24.5	0.677
		Defenders	3.05	1.47	81.1	0.056
		Midfielders	2.16	1.47	9.8	0.462
Anterioromedial Stability	Goalkeepers	Defenders	3.64	1.43	62.4	0.032 *
		Midfielders	2.34	1.43	66.5	0.024 *
		Attackers	1.44	1.43	72.1	0.745
	Defenders	Goalkeeper	−3.64	1.43	62.4	0.032 *
		Midfielders	−1.29	1.43	43.2	0.802
		Attackers	−2.19	1.43	42.7	0.425
	Midfielders	Goalkeeper	−2.34	1.43	66.8	0.024 *
		Defenders	1.29	1.43	43.2	0.802
		Attackers	−0.89	1.43	35.2	0.924
	Attackers	Goalkeepers	−1.44	1.43	72.1	0.745
		Defenders	2.19	1.43	42.7	0.425
		Midfielders	0.89	1.43	35.2	0.924

* Significant at 0.05 level.

**Table 6 healthcare-10-02256-t006:** Statistical analysis of variance of lower limb strength and power values of soccer players for their playing position.

Dependent Variable (cm)	Groups	Mean ± Std	95% CI Lower to Upper Limits	Partial Eta- Squared	*p* Value
Lower limb strength and power	Goalkeeper	565.05 ± 72.70	534.36–595.75	0.421	0.008 *
	Defenders	508.55 ± 58.19	483.98–533.12		
	Midfielders	482.16 ± 93.91	442.50–521.82		
	Attackers	526.18 ± 99.69	484.09–568.27		

* Significant at 0.05 level.

**Table 7 healthcare-10-02256-t007:** Post-Hoc (Tukey) analysis of lower limb strength and power values of soccer players for their playing position.

Dependent Variable (cm)	Group	Group	Mean Difference	Std. Error	Effect Size (%)	*p* Value
Lower limb strength and power	Goalkeepers	Defenders	56.51	23.90	84.4	0.091
		Midfielders	82.89 *	23.90	92.8	0.004 *
		Attackers	38.88	23.90	33.8	0.369
	Defenders	Goalkeeper	−56.51	23.90	84.4	0.091
		Midfielders	26.39	23.90	21.5	0.688
		Attackers	−17.63	23.90	11.3	0.882
	Midfielders	Goalkeeper	−82.89 *	23.90	92.8	0.004 *
		Defenders	−26.39	23.90	21.5	0.688
		Attackers	−44.02	23.90	35.2	0.261
	Attackers	Goalkeepers	−38.88	23.90	33.8	0.369
		Defenders	17.63	23.90	11.3	0.882
		Midfielders	44.02	23.90	35.2	0.261

* Significant at 0.05 level.

## Data Availability

Data of this study is secure with the first author and available on request.

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
