# Peer review of "Analysis of the Anaerobic Power Output, Dynamic Stability, Lower Limb Strength, and Power of Elite Soccer Players Based on Their Field Position"

_healthcare, 2022, doi:10.3390/healthcare10112256_

Round 1
Reviewer 1 Report
Thank you for requesting a peer review.
I have many concerns about background and some results, etc.
Please consider the following comments.
Which is the novelty of this study “recognizing physical performance among 14 elite soccer players based on their field position” or “investigating in the Middle East region”?
Previous studies have covered soccer players according to their playing position [15–19] and investigated their physical characteristics [20].
If conducting examinations in the Middle East area is novelty, you should state the reason and need for the research in this area. What are the reasons why the results of an international study cannot be reflected in the Middle East area? (Performance level? Musculoskeletal differences?)
Why you stated countermovement vertical jump in background section, although you selected SLVJ as anaerobic power output. Consistency is needed.
Line 151: he → they?
I recommend adding a graphic figure related the SEBT. If possible, other tests as well.
Line 180: p<0.05?
Line 191: A total…. It is a duplicate of Methods (line 110-112). I recommend integrating it into the methods.
Line 198-203: Are these results the results of post hoc? If so, please state p-value.
Table 4 : please remove N.
Table 4 : Please show Mean and Std (cm) not Deviation.
Why was the dynamic stability measured by SEBT not normalized their height or Leg length?  The effect of height on the results must be compensated for.
There is a lot of overlap between the description of the results and the table contents. I recommend that you organize what you present in the tables and descriptions.
Author Response
Comments and Suggestions for Authors
Thank you for requesting a peer review.
I have many concerns about background and some results, etc.
Please consider the following comments.
Which is the novelty of this study “recognizing physical performance among 14 elite soccer players based on their field position” or “investigating in the Middle East region”?
Even though there are few studies investigated some of the physical characteristics of soccer players in the middle east region, non-of them investigated the most important soccer performance characteristics such as anaerobic power, dynamic stability, and strength and power of lower leg muscles based on the field position in this region. Moreover, this is the first study which investigated differences in performance characteristics based on field position among Saudi soccer players.
Previous studies have covered soccer players according to their playing position [15–19] and investigated their physical characteristics [20].
If conducting examinations in the Middle East area is novelty, you should state the reason and need for the research in this area. What are the reasons why the results of an international study cannot be reflected in the Middle East area? (Performance level? Musculoskeletal differences?)
Ethnicity and genetics have a major role in determining athletes' physical, and physiological characteristics and performance. It has been reported in previous studies that the athletes from the middle east region differ from other parts of the world in physical and anthropometric characteristics.
Why you stated countermovement vertical jump in background section, although you selected SLVJ as anaerobic power output. Consistency is needed.
countermovement vertical jump changed as single leg vertical jump to maintain consistency.
Line 151: he → they?
They- Changed
I recommend adding a graphic figure related the SEBT. If possible, other tests as well.
Line 180: p<0.05?
The Normality and homogeneity of the data confirmed if p<0.05.
Line 191: A total…. It is a duplicate of Methods (line 110-112). I recommend integrating it into the methods.
Removed from the line 191.
Removed from the line 191
Line 198-203: Are these results the results of post hoc? If so, please state p-value.
P-value added
Table 4 : please remove N.
Removed
Table 4 : Please show Mean and Std (cm) not Deviation.
Changed
Why was the dynamic stability measured by SEBT not normalized their height or Leg length?  The effect of height on the results must be compensated for.
We did not do the normalized with their height or leg length and this was the study's limitation
There is a lot of overlap between the description of the results and the table contents. I recommend that you organize what you present in the tables and descriptions.
Done
Reviewer 2 Report
To improve the paper I strongly suggest to add the data about reliability, ICC or similar about all test used.
1) it'is important to declare the manufactory of Vertec device
2) how the authors verify and control teh 90 deg angle at the knee before the jump?
3) how is the repeatability of SEBT test? please provide the ICC o other reference
4) how is the repeatability of THD test? please provide the ICC o other reference
5) in all role of player the author recruited perfectely, 24 player. so the author have to define a 'the sample as 'convenince sample' and erase the repetition of 24 in the table...because is always 24
6) the test in 2.6.1 is not anaerobic power output but only power output
7) the test in 2.6.2 is dynamic balance and not stability
8) table 2 and 6 could be graphs
Author Response
Comments and Suggestions for Authors
To improve the paper, I strongly suggest to add the data about reliability, ICC or similar about all test used.
1) it's important to declare the manufactory of Vertec device
Vertec device (Jump USA 22550).
2) how the authors verify and control the 90 deg angle at the knee before the jump?
We used a manual goniometer to measure the knee flexion angle at 900. Before actual jump participant flex his knee up to 90 deg angel. He makes 2- 3 trails to bend his knee up to 90 deg angle.
3) how is the repeatability of SEBT test? please provide the ICC o other reference
The SEBT's intraratar and interrater reliability (ICC) values for all directions ranged from 0.88 to 0.96, and from 0.83 to 0.93 respectively [27].
4) how is the repeatability of THD test? please provide the ICC o other reference
The test-retest reliability of THD has been demonstrated as intraclass correlation coefficient = 0.98 in previous research [29].
5) in all role of player the author recruited perfectly, 24 player. so the author have to define a 'the sample as 'convenience sample' and erase the repetition of 24 in the table...because is always 24
Done (erase)
6) the test in 2.6.1 is not anaerobic power output but only power output
We can measure the anaerobic power as well, references provided below…
https://pubmed.ncbi.nlm.nih.gov/21186549/
http://www.doiserbia.nb.rs/Article.aspx?id=0025-81051006371O#.Yzv1ZnZByUk
7) the test in 2.6.2 is dynamic balance and not stability
Dynamic balance and dynamic stability are interchange terms. We can use anyone. There are two references as they used as to measured dynamic stability.
https://www.sciencedirect.com/science/article/abs/pii/S0966636215008978
https://www.ncbi.nlm.nih.gov/pmc/articles/PMC3396295/
8) table 2 and 6 could be graphs
As there are values like CI and effect size that could not be represent by the graphs, we feel that its better to be in the table.

Round 2
Reviewer 1 Report
I think most of it has been revised. Thank you very much.
However, there are a few point that should be revised, please see below.
---
Line 191: Is p<0.05 correct, not p>0.05?
Table 4. Are ”the Mean±Std” results correct? Does these data show Deviation?
I think it needs to be rewritten to cm data.
You did not do the normalized with their height or leg length and this was the study's limitation. This could be a major problem. It is recommended that the reasons why normalization could not be done be included.
Author Response
"Please see the attachment"
